

# Pan-Antarctic map of near-surface permafrost temperatures at 1 km² scale

Jaroslav Obu[1], Sebastian Westermann[1], Gonçalo Vieira[2], Andrey Abramov[3], Megan Balks[4], Annett Bartsch[5,6], Filip Hrbáček[7], Andreas Kääb[1], Miguel Ramos[8]

[1]Department of Geosciences , University of Oslo, Oslo, Sem Sælands vei 1, 0371 Oslo, Norway
[2]Centre of Geopgraphical Studies, Institute of Geography and Spatial Planning, University of Lisbon, R. Branca Edmée Marques, 1600-276 Lisbon, Portugal
[3]Institute of Physicochemical and Biological Problems in Soil Science, Russian Academy of Sciences, Pushchino, Russia
[4]Department of Earth and Ocean Sciences, University of Waikato, Hamilton, Private Bag 3105, New Zealand
[5]Zentralanstalt für Meteorologie und Geodynamik, Hohe Warte 39, Vienna, Austria
[6]now at b.geos GmbH, Vienna, Industriestrasse 1 2100 Korneuburg, Austria
[7]Department of Geography, Masaryk University, Kotlářská 2 61137 Brno, Czech Republic
[8]Department of Physics and Mathematics, University of Alcalá, Madrid, Campus universitario 28805 Alcalá de Henares, Spain

*Correspondence to*: Jaroslav Obu (jaroslav.obu@geo.uio.no)

## Abstract

Permafrost is present under almost all of the Antarctic's ice-free areas but little is known about spatial variations of permafrost temperatures outside a few areas with established ground temperature measurements. We modelled a temperature at the top of the permafrost (TTOP) for all the ice-free areas of Antarctic mainland and Antarctic Islands at 1 km² resolution during 2000–2017. The model was driven by remotely-sensed land surface temperatures and down-scaled ERA-Interim climate reanalysis data and subgrid permafrost variability was simulated by variable snow cover. The results were validated against in-situ measured ground temperatures from 40 permafrost boreholes and the resulting root mean square error was 1.9 °C. The lowest near-surface permafrost temperature of -33 °C was modelled at Mount Markham in Queen Elizabeth Range in the Transantarctic Mountains. This is the lowest permafrost temperature on Earth according to the modelling results on global scale. The temperatures were most commonly modelled between -23 and -18 °C for mountainous areas rising above the Antarctic Ice Sheet and between -14 and -8 °C for coastal areas. The model performance was good where snow conditions were modelled realistically but errors of up to 4 °C can occur at sites with strong wind-driven redistribution of snow.



## 1 Introduction

Permafrost in the Antarctic is present beneath all ice-free terrain, except for the lowest elevations of the maritime Antarctic and sub-Antarctic islands (Vieira et al., 2010), unlike the northern hemisphere, where permafrost underlies approximately 15 % of the unglaciated land. The ice and snow-free land occupies 0.22% (30 900 km$^2$) of the whole Antarctic (Burton-Johnson

et al., 2016; Hrbáček et al., 2018). The major ice-free areas include the Queen Maud Land, Enderby Land, the Vestfold Hills, Wilkes Land, the Transantarctic Mountains, the Ellsworth Mountains, Marie Byrd Land and the Antarctic Peninsula (Green et al., 1967; Fig. 1). Despite the relatively small area, in comparison to glaciated areas, permafrost is one of the major factors controlling terrestrial ecosystem dynamics in the Antarctic (Bockheim et al., 2008).

Compared with the Northern Hemisphere, where the first permafrost investigations date back to 19th century (Shiklomanov, 2005; Humlum et al., 2016), the ground temperatures in the Antarctic have been systematically studied only during the last two decades. Permafrost was studied in relation to patterned ground since the 1960s and during the Dry Valley Drilling Project in the 1970s but temperatures have been measured only occasionally (Decker and Bucher, 1977; Guglielmin, 2012). A first Antarctic permafrost borehole network was implemented in 1999 in the Victoria Land (Transantarctic Mountains) and was

extended during the International Polar Year 2007–2009 to cover all eight major ice-free regions (Vieira, 2010).

Permafrost distribution was estimated and mapped on the Antarctic Peninsula by Bockheim et al. (2013) based on mean annual temperature, periglacial features, shallow excavations, borehole measurements, geophysical surveys, and existing permafrost models. Bockheim et al. (2007) characterised permafrost in the McMurdo Dry Valleys based on ground ice properties and

active layer thickness from more than 800 shallow excavations.

Antarctic permafrost modelling efforts were limited to small areas in the Antarctic Peninsula region and sub-Antarctic Islands. Ferreira et al. (2017) used freezing indexes and the temperature at the top of the permafrost (TTOP) modelling for eight monitored sites on Hurd Peninsula, Livingston Island for the 2007 and 2009 seasons to study the controlling factors of ground

temperatures. Rocha et al., (2010) ran the H-TESSEL scheme forced by ERA-Interim reanalysis to simulate ground temperatures at Reina Sofia Peak on Livingston Island. Ground temperature measurements and permafrost modelling efforts have been limited to point sites and little is known about spatial variability of ground temperatures at the regional and continent-wide scales.

In this study we employed the TTOP modelling scheme based on The Moderate Resolution Imaging Spectroradiometers (MODIS) land surface temperatures (LST) and ERA-Interim reanalysis to model the spatial distribution of temperatures at the top of permafrost on all ice-free areas of Antarctic and Antarctic Islands. We adapt the existing modelling scheme from the



Northern Hemisphere (Westermann et al., 2015; Obu et al., 2019) according to the available input data and their characteristics for the Antarctic.

Figure 1: Overview map of the ice-free Antarctic regions and extents of the maps presented in the paper.



## 2 Methods

### 2.1 The Cryogrid 1 model

The CryoGrid 1 model (Gisnås et al., 2013) calculates mean annual ground temperature (MAGT) and is based on the TTOP equilibrium approach (Smith and Riseborough, 1996):

$$
MAGT = \begin{cases} \frac{1}{\tau}\left(n_f FDD + r_k n_t TDD\right) \\ for\ \left(n_f\ FDD + r_k n_t TDD\right)\ \leq 0 \\ \frac{1}{\tau}\left(\frac{1}{r_k} n_f FDD + n_t TDD\right) \\ for\ \left(n_f\ FDD + r_k n_t TDD\right)\ > 0 \end{cases}
$$

Where: FDD and TDD represent freezing and thawing degree days, respectively, of the surface meteorological forcing
accumulated over the model period $\tau$ in days. The influence of the seasonal snow cover, vegetation, and ground thermal properties were taken into account by the semi-empirical adjustment factors $r_k$ (ratio of thermal conductivity of the active layer in thawed and frozen state), $n_f$ (scaling factor between average winter surface and ground surface temperature) and $n_t$ (scaling factor between average summer surface and ground surface temperature). We use land surface temperature (LST) to compute FDD and TDD (see following section) instead of the air temperature that was used initially by Smith and Riseborough (2002)
and hence omit thawing $n_t$-factors.

### 2.2 Freezing and thawing degree days

Spatially distributed data sets of TDD and FDD were compiled from remotely sensed land surface temperature products and climate reanalysis data following the procedure of Obu et al. (2019). We used LST data (level 3 product in processing version
6) from MODIS on board the Terra and Aqua satellites, which contain up to two daytime and two night-time measurements per day at a spatial resolution of 1 km since the year 2000 (Wan, 2014). For this reason, the study extends from 2000 and to the end of 2017. Data gaps in the MODIS LST time series due to cloud cover can result in a systematic cold bias in seasonal averages (Westerman et al., 2012; Soliman et al., 2012; Østby et al., 2014), so a gap filling with near-surface air temperatures from the ERA-interim and ERA-5 reanalysis was applied (Westermann et al., 2015).

The ERA-interim reanalysis provides gap-free meteorological data from 1979 onwards at a spatial resolution of 0.75° × 0.75° (Dee et al., 2011). The ERA-5 reanalysis is ERA-Interim upgrade and provides the data at improved 0.28125° × 0. 0.28125° (31 km) spatial resolution but was at the time of the study available only from 2008 onwards (Hersbach and Dee, 2016). ERA-





Interim data were for this reason used before 2008 and ERA-5 data afterwards. The reanalysis data were downscaled to the 1 km resolution of individual MODIS pixels using atmospheric lapse rates and The Global Multi-resolution Terrain Elevation Data 2010 (GMTED2010) (Danielson and Gesch, 2011). The downscaling methodology is described in detail by Fiddes and Gruber (2014), Westermann et al. (2015) and Obu et al. (2019). The gap-filled MODIS LST time series were averaged to eight-day periods from which FDD and TDD were finally accumulated for the 2000–2017 study period.

## 2.3 Average annual snowfall and $n_f$-factors

Spatially distributed data sets of $n_f$-factors were generated from average annual snowfall forced by ERA-Interim and ERA-5 reanalysis data. Snow cover in the majority of the Antarctic is dominated by sublimation and processes related to blowing snow (Gallée, 1998; Bintanja and Reijmer, 2001), which were not taken in to account by the snowfall and degree-day model that Obu et al. (2019) used to estimate $n_f$-factor for the Northern Hemisphere. For this reason, only mean annual snowfall was calculated using downscaled ERA-Interim precipitation before 2008 and ERA-5 precipitation after. Precipitation was downscaled based on the difference between reanalysis elevation and GMTED2010 using a precipitation gradient, found in drier areas (Hevesi et al., 1992), of 2 % per 100 m up to 1000 m and 1% per 100 m for elevations above 1000 m. Snowfall was defined as precipitation at air temperatures below 0 °C, using the downscaled ERA-Interim air temperatures as employed for the gap filling of MODIS LST (see above). For a detailed downscaling procedure description see Obu et al. (2019).

$N_f$-factors were defined based on average annual snowfall, however the reported $n_f$-factors in Antarctic were calculated in respect to snow depth. Oliva et al. (2017) identified $n_f$-factor values around 0.3 for snow accumulations of 80 cm on Livingston Island, although the $n_f$-factors can reach up to 0.55 in the thick snow cover due to its temporal variability (de Pablo et al., 2017). $N_f$-factors close to, or even greater than, one were measured in areas with little or no snow cover on Veleskarvet nunatak (Kotzé and Meiklejohn, 2017), in the McMurdo Dry Valleys (Lacelle et al., 2016) and on James Ross Island (Hrbáček et al., 2016). This range was used to constrain the $n_f$-factor ranges in relation to average annual snowfall (Table 1). Since the snow model was not able to simulate snow free-sites nor snowdrifts due to strong wind distribution, we used a maximum $n_f$-factor of 0.95 and minimum of 0.3. However, smaller snow-depth variations on a local scale were taken into account with an ensemble of different values of average annual snowfall for each 1 km$^2$ pixel (e.g. Gisnås et al., 2014).



| Mean annual snowfall (mm) | $n_f$ min | $n_f$ max |
|---|---|---|
| < 3 | 0.85 | 0.95 |
| 3-10 | 0.77 | 0.85 |
| 10-30 | 0.75 | 0.77 |
| 30-50 | 0.73 | 0.75 |
| 50-75 | 0.67 | 0.73 |
| 75-100 | 0.64 | 0.67 |
| 100-125 | 0.55 | 0.64 |
| 125-150 | 0.5 | 0.55 |
| 150-200 | 0.45 | 0.5 |
| 200-300 | 0.4 | 0.45 |
| > 300 | 0.3 | 0.4 |

**Table 1: ranges of $n_f$-factors that were assigned to mean annual snowfall values.**

**2.4 $r_k$-factors**

The $r_k$-factor is defined as the ratio of thawed and frozen thermal conductivities of the active layer material (Romanovsky and Osterkamp, 1995) and is related to water and organic contents (e.g. Gisnås et al., 2013). Soil moisture properties were mapped on a regional scale (Bockheim et al., 2007) but no pan-Antarctic datasets related to soil water or organic contents were available. The ESA CCI Landcover that Obu et al. (2019) used for Northern Hemisphere contains only "permanent snow and ice" class on the Antarctic mainland, therefore, a rock outcrops dataset (Burton-Johnson et al., 2016) was used to constrain

non-glaciated areas. An $r_k$-factor of 0.85 was used for the whole Antarctic, representing an average value between very dry sites on continental Antarctic and moderately moist sites on the Antarctic Peninsula.

**2.5 Ensemble-based modelling of subpixel heterogeneity**

Ground temperatures can vary considerably at short distances due to heterogeneous snow cover, vegetation, topography and

soil properties (Beer, 2016; Gisnås et al., 2014; 2016; Zhang et al., 2014). We ran an ensemble of 200 model realisations with different combinations of $n_f$- and $r_k$-factors to simulate the variability. $R_k$-factor values were set to randomly vary by $\pm 0.1$ between 0.75 and 0.95 to represent both very dry sites and locations with higher soil moisture. The distribution of snowfall within the 1 km pixel was simulated using a log-normal distribution function where mean annual snowfall determined the mean of the distribution. The coefficient of variation of the distribution for the open areas (0.9) according to Liston (2004) was

assigned to all modelled areas. An $n_f$-factor was assigned to the estimated average annual snowfall according to Table 1. The



pixels not overlapping with rock outcrops were masked out. A fraction of model runs with MAGT < 0 °C was used to derive permafrost type (zones) on Antarctic Islands (see Obu et al. (2019) for detailed description).

## 2.6 Model validation

We compared our results to available in-situ measurements in 40 permafrost boreholes and shallow boreholes at soil-climate stations. The ensemble mean of modelled MAGTs was compared to the borehole measurements to take the simulated spatial variability, provided by the ensemble spread, into account. The accuracy of the model was estimated with root mean square error (RMSE) and mean absolute error between the modelled and measured MAGT.

The validation data were provided by the authors for the McMurdo Dry Valleys, ice-free areas near Russian stations (Bunger Hills, Schirmacher Hills, Larsemann Hills, Thala Hills, King George Island and Hobs coast) and the Northern Antarctic Peninsula. Ground temperatures from these locations represent mean MAGT at the top of the permafrost and usually overlap well with the modelling period 2000–2017 (Appendix A). Validation data for Queen Maud Land (Troll Station, Flarjuven Bluff, Vesleskarvet) and the Baker Rocks site were obtained from Hrbáček et al., (2018) and MAGTs from Terra Nova Bay (Oasi New and Boulder Clay) were obtained from Vieira et al., (2010). MAGTs from Hope Bay, Mt. Dolence, Marble Point Borehole and Limnopolar Lake were obtained from Schaefer et al. (2017a), Schaefer et al. (2017b), Guglielmin et al. (2011) and de Pablo et al. (2014), respectively. The borehole data from Signy Island and Rothera Point were extracted from Guglielmin et al., (2012) and (2014). The data found in publications were not necessarily calculated for the top of the permafrost and do not completely overlap with the modelling period and are, thus, less reliable than the author-supplied validation data. For instance, the data from Vieira et al. (2010) represents MAGT for the periods before 2010 usually lasting only few years (Table 2).

## 3 Results

### 3.1 Comparison to borehole measurements

Ground temperatures can vary significantly inside a 1 km² model pixel, which is, to a certain extent, represented by the TTOP model ensemble runs. Average MAGT derived from the ensemble runs was compared to the measured site ground temperatures, which might limit representativeness for sites with locally specific ground and snow properties. The comparison yielded a RMSE of 1.94 °C and a mean absolute error for all boreholes of 0.06 °C (Fig. 2). The small mean absolute error is partly achieved with fine adjustment of the $n_f$-factor class limits. For 50 % of the boreholes, the agreement between borehole temperatures and modelled MAGT was better than 1 °C, while it is better than 2 °C for 75%, and better than 3 °C for 85 % of the boreholes. Assuming a Gaussian distribution of standard deviation σMAGT, 68% of borehole comparisons should fall within one σMAGT, while 95% within two σMAGT, and 99% should be within three σMAGT. For the comparison with



Antarctic boreholes, 18 (45 %) boreholes were contained within one, 29 (73 %) within two, and 31 (78 %) within three standard deviations from the mean, which is comparable to the results for the Northern Hemisphere (Obu et al., 2019).

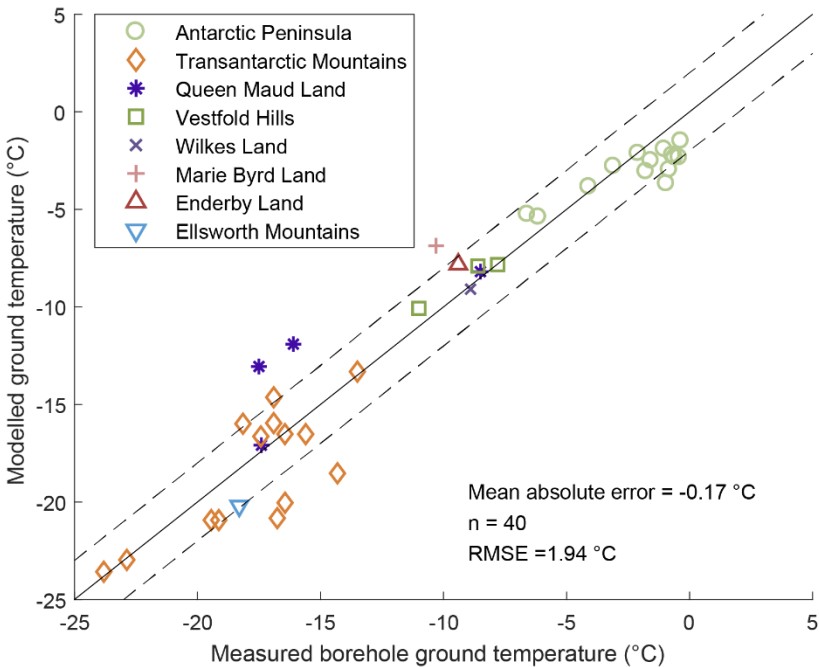

**Figure 2: Measured vs. modelled permafrost temperatures for all boreholes. The dashed lines represent ±2 °C intervals around the 1:1 solid line.**

## 3.2 Queen Maud Land

There are 2430 km² of ice free areas in Queen Maud Land according to the rock outcrop map (Burton-Johnson et al., 2016; Hrbáček et al., 2018). The average MAGT is -18.2 °C and it ranges from -26.2 °C in the highest parts of Fimbulheimen Range to -6.3 °C in Prince Olav Coast, where MAGTs down to -10 °C were modelled at elevations exceeding 200 m (Fig. 3). MAGTs above -10 °C can be found also at Schirmacher Oasis (Hills) and reach above -8 °C. MAGTs in Sør Rondane Mountains and in Fimbulheimen Range range from -12 °C at elevations of around 800 m a.s.l. to -24 °C at elevations exceeding 3000 m a.s.l.. Kirwan Escarpment (elevations usually exceeding 2000 m) is characterised by MAGTS between -23 °C and -20 °C and in Heimefront Range (with slightly lower elevations) MAGTs were between -22 °C and -18 °C. MAGTs in Borg Massif weref modelled from -21 to -17 °C and between -16 and -12 °C in Ahlmann Ridge.



**Figure 3: Permafrost temperature maps of Queen Maud Land and differences between borehole and modelled MAGT. See Fig. 1 for location of panels (a) and (b).**



### 3.3 Enderby Land

Permafrost occupies 1140 km² of ice-free area in Enderby Land, which is predominantly mountain tops and a few coastal sites. The modelled average MAGT was -11.7 °C, ranging from -22.4 °C, on summits exceeding 2000 m elevation, to -6.3 °C in the northwest part of the coast (Fig. 4). The MAGT was modelled as around - 8 °C at the majority of the coast and around -10 °C
5  in the coastal areas of the Nye, Scott and Tula Mountains, dropping below -15 °C at elevations exceeding 1000 m a.s.l. In the Framnes Mountains, MAGT was modelled at between -17 and -12 °C.





**Figure 4: Permafrost temperature maps of Enderby Land and differences between borehole and modelled MAGT. See Fig. 1 for location of panels (a) and (b).**

## 3.4 Vestfold Hills

5   The ice-free area of the Vestfold Hills region is 2750 km². The modelling showed an average MAGT of -17.4 °C in this region. The MAGT ranged from -6.6 °C on the islands of the Ingrid Christensen Coast to -28.3 °C in the highest parts of the Prince Charles Mountains (Fig. 5). The MAGT in Amery Oasis, which is a part of the Prince Charles Mountains, was modelled as -13 °C, in the lowest-lying areas, down to -18 °C at elevations approaching 1000 m a.s.l. At the similar elevations on the Mawson Escarpment significantly lower MAGTs, from -19 °C at 200 m a.s.l down to -24 °C at 1500 m a.s.l were recognised.

10  In the coastal lowland areas of the Larsemann and Vestfold Hills the MAGT ranged between -10 to -7 °C.

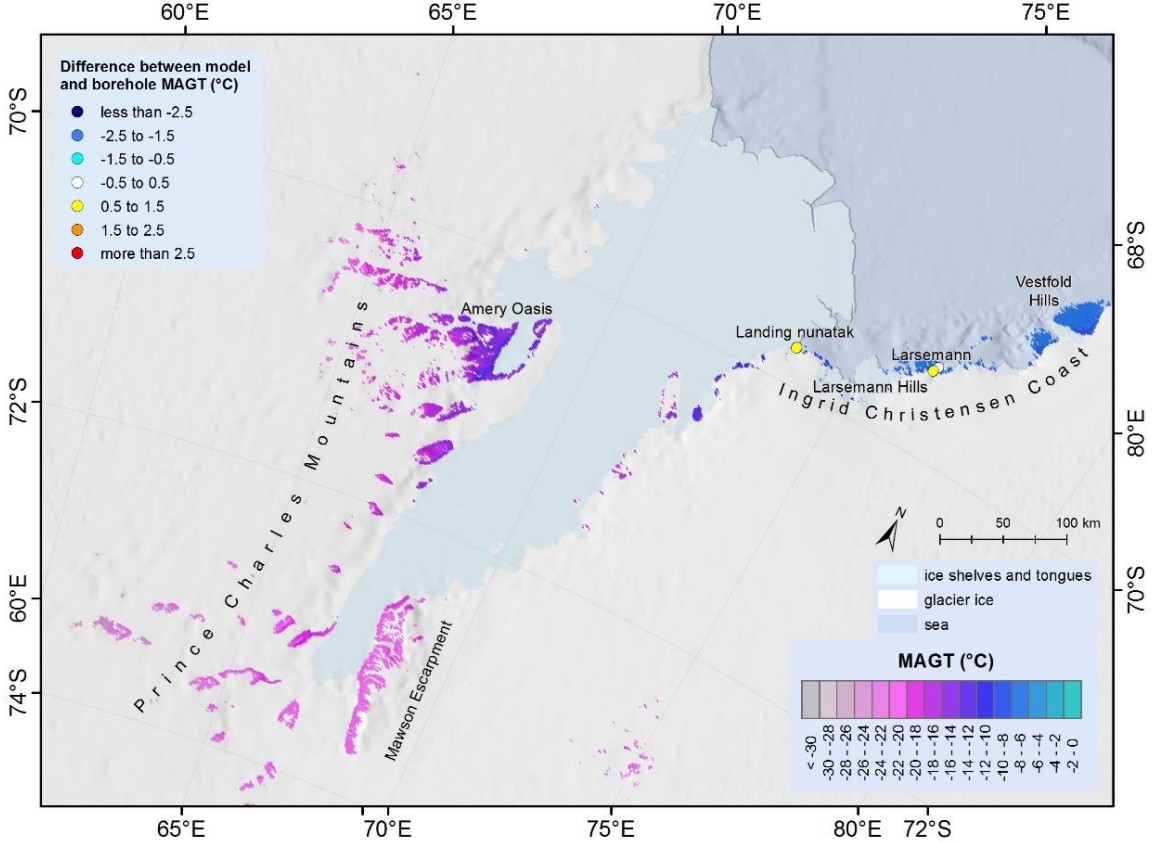

**Figure 5: Permafrost temperature map of the Vestfold Hills and differences between borehole and modelled MAGT.**





## 3.5 Wilkes Land

The majority of the 400 km² of ice free area in Wilkes Land lies in the area surrounding the Bunger Hills, where MAGTs of around -9°C were modelled close to the Shackleton Ice Shelf. The lowest MAGT of -15.9 °C was modelled on the adjacent mountains at 1300 m elevation (Fig. 6). Modelled MAGTs were -8 to -6 °C at the Budd Coast, -8 °C at the Adélie Coast, and
5  -10 to -11 °C at the George V Coast. Due to the prevalence of low-lying regions the mean MAGT of the region was only -8.9 °C.

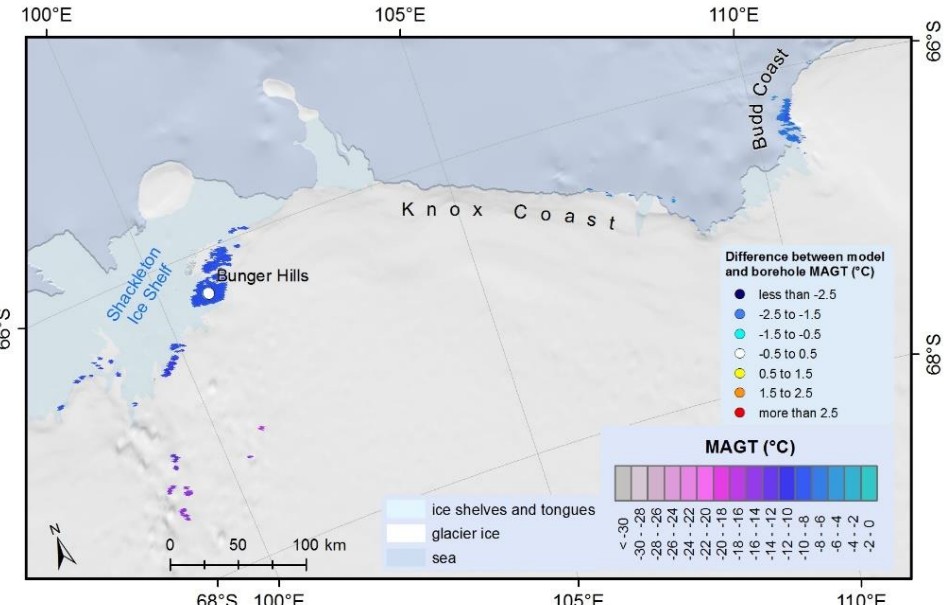

**Figure 6: Permafrost temperature maps of Wilkes Land and differences between borehole and modelled MAGT. See Fig. 1 for**
10 **location.**

## 3.6 Transantarctic Mountains

The Transantarctic Mountains are the largest ice-free region, comprising 19 750 km² and extending from Cape Adare to Coats Land. The part west of the 90° meridian (Fig. 7) consists of mountain ranges mostly lower than 2000 m (except for the Thiel Mountains) that don't extend to the sea level. The highest MAGT of -17.0 °C was modelled at the foot of the Shackleton Range
15 and the Pensacola Mountains. The MAGTs decreased down to -29 °C in the high mountains. The lowest MAGT of -29.8 °C was modelled in the Thiel Mountains.



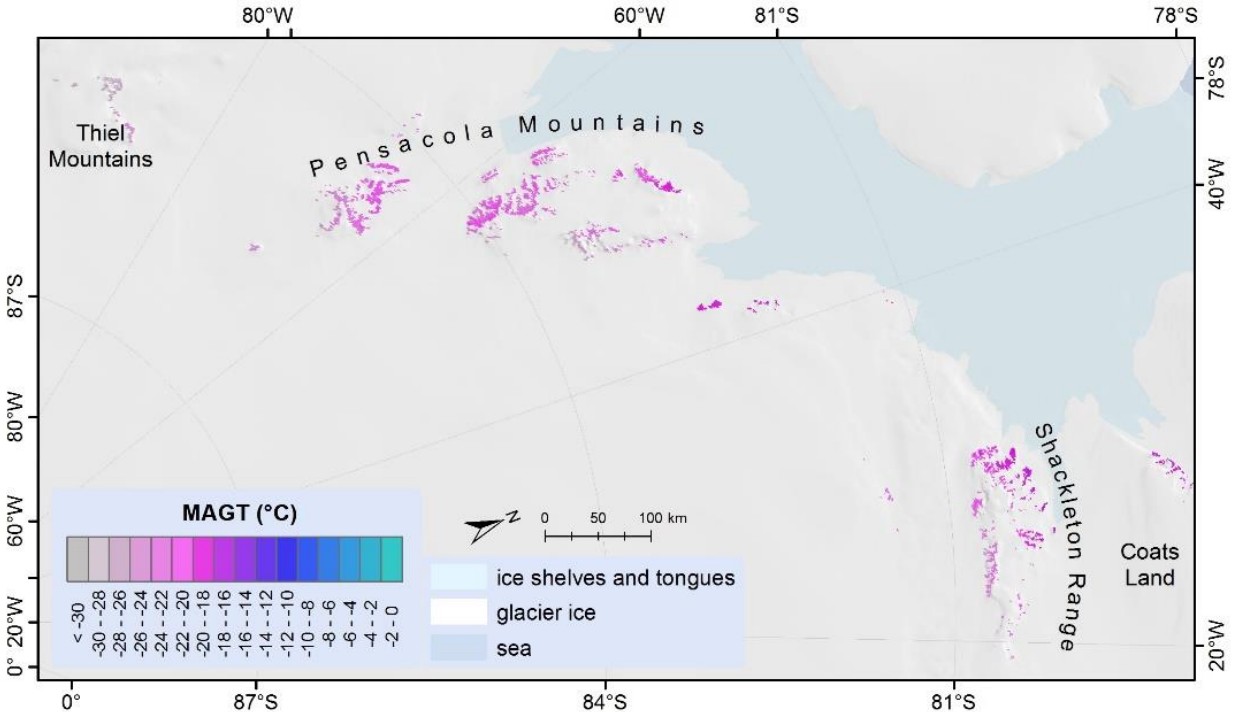

**Figure 7: Permafrost temperature map of the Transantarctic Mountains west of the 90° meridian. See Fig. 1 for location.**

The region east of the 90° meridian (Fig. 8) consists of numerous mountain ranges extending from the Ross Ice Shelf and Ross
Sea up to more than 4000 m elevation and from 85° S to 69° S latitude. The results show the widest range of MAGTs among
all the regions with the lowest temperature of -33.5 °C at Mount Markham in Queen Elizabeth Range to -8.5 °C on the Oates
Coast. The modelled MAGTs close to the Ross Ice Shelf decreased northwards from -15 °C at the Amundsen Coast to -18 °C
at the north of the Dufek Coast. Similar MAGTs, of between -19 and -17 °C were modelled further north along the ice shelf
at the Shackleton and Hillary Coasts. Similar decreases in MAGT at higher elevations in the northward direction were
observed, but local MAGT variations in relation to altitude were considerable. However, MAGTs below - 30 °C were modelled
at the highest parts of the mountain ranges along the Ross Ice Shelf. Higher MAGTs were modelled along the Ross Sea and
range between -15 °C and -12 °C along the Scott and Borchgrevink Coasts. The modelled MAGTs were around -10 °C at Cape
Adare and between -13 and -10 °C on the Pennell Coast. MAGT was modelled down to -26 °C in the Prince Albert Mountains
at elevations above 2000 m a.s.l. They approach -30 °C at the highest elevations, that exceed 3000 m a.s.l., in the Deep Freeze
Range. Despite the elevations reaching 4000 m a.s.l in the Admiralty Mountains, the MAGT was only down to -23 °C.



**Figure 8: Permafrost temperature maps of the Transantarctic Mountains east of the 90° meridian and differences between borehole and modelled MAGT. See Fig. 1 for location of panels (a), (b) and (c).**





### 3.6.1 McMurdo Dry Valleys

The McMurdo Dry Valleys are a part of Transantarctic Mountains and include a large ice-free area (6700 km$^2$) and are one of the most extensively studied permafrost regions in Antarctica (Bockheim et al., 2007). The lowest MAGT among the dry valleys is modelled in the Victoria Valley falling below -24 °C at the lowest part (Fig. 9). A winter ground temperature inversion is pronounced with MAGTs in the surrounding valleys modelled at around -21 °C. The MAGT also increases up the valleys to -21 °C in the McKelvey, Balham and Barwick Valleys. No MAGT inversion was modelled in the Wright Valley, where MAGTs ranged between -21 and -19 °C and in the Taylor Valley, which is the warmest with MAGT of -17 °C in the lower-lying parts, and -20 °C in the upper part, of the valley. At the Olympus and Asgard Ranges and the Kukri Hills, which surround the valleys, MAGTs of between -23 and – 20°C were modelled. In the surrounding mountains, close to the ice sheet, modelled MAGTs were around -25 °C and reached -27 °C at the highest and the most east-lying mountains. MAGTs at the coast of McMurdo Sound ranged from -13 to -17 °C. On Ross Island, the MAGT was modelled to -16 °C at the Scott and McMurdo Bases which are near sea level and down to -24 °C on Mount Erebus.



**Figure 9: Permafrost temperature map of the McMurdo Dry Valleys and differences between borehole and modelled MAGT. See Fig. 1 for location. Note: the MAGT colour ramp is different from other figures. 1: McKelvey Valley; 2: Balham Valley; 3: Barwick Valley; 4: Olympus Range; 5: Asgard Range 6: Kukri Hills.**

## 3.7 Ellsworth Mountains

The ice-free area of the Ellsworth Mountains occupies 380 km$^2$ of high-elevation terrain; therefore the modelled mean MAGT was -21.5 °C. The highest temperature was modelled at the foot of the mountains (-17.4 °C) at 500 m a.s.l. with -21 °C at 1000 m a.s.l., -22 °C at 2000 m, and down to -26.1 °C at the highest elevations of Vinson Massif (Fig. 10).



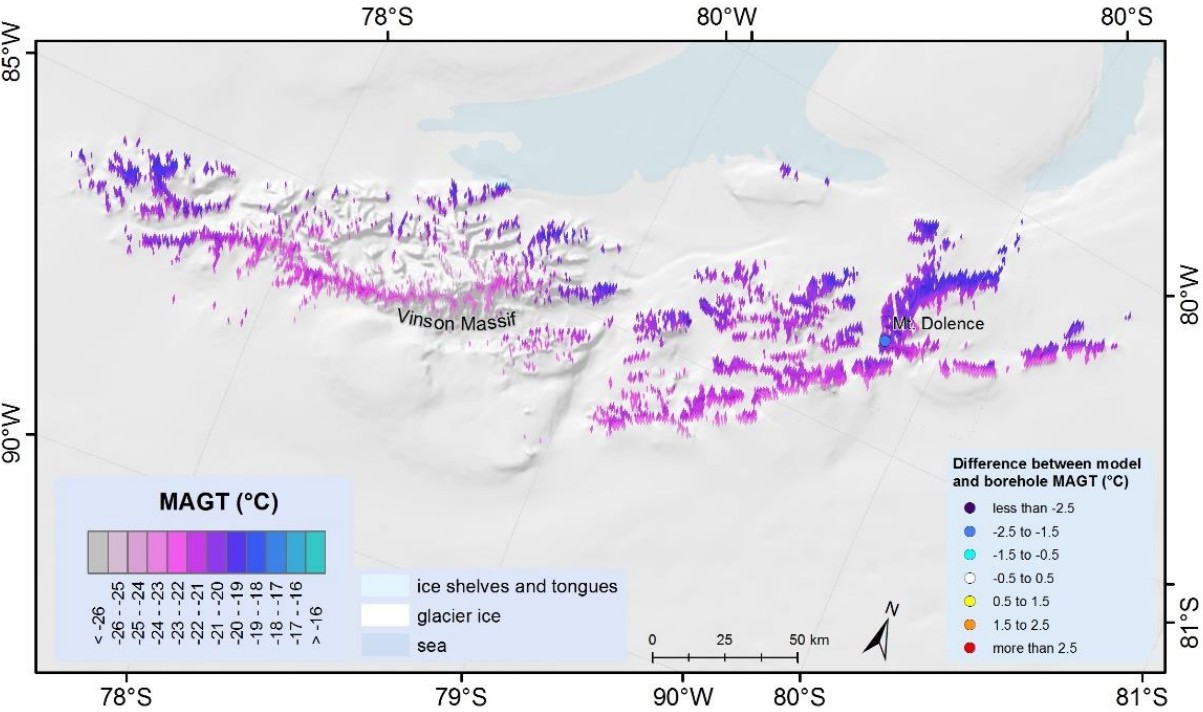

**Figure 10: Permafrost temperature map of the Ellsworth Mountains and differences between borehole and modelled MAGT. See Fig. 1 for location.**

## 3.8 Marie Byrd Land

5    The ice free areas in Marie Byrd Land occupy only 210 km² and consist mostly of rock outcrops at lower elevations close to the coast and volcanos protruding through the ice sheet. In the Ford Ranges, MAGT was modelled as from -10 °C close to sea level to -16 °C at elevations exceeding 1000 m a.s.l (Fig. 11). MAGTs between -11 and -8 °C were modelled at the Hobbs, Walgreen, Eights and Ruppert Coasts, reaching -6 °C on the islands surrounded by open sea. The volcano-mountain ranges reaching 3000 m a.s.l. (Flood and Kohler Range), had MAGTs typically ranging between -16 to -14 °C at their peaks. Modelled

10    MAGTs were between -25 and -21 °C on the Executive Committee Range, where rock outcrops occur above 2000 m a.s.l. and the highest peaks extend to over 4000 m a.s.l.



**Figure 11: Permafrost temperature maps of Marie Byrd Land and differences between borehole and modelled MAGT. See Fig. 1 for location of panels (a) and (b).**

5   **3.9 Antarctic Peninsula**

The ice-free areas of the Antarctic Peninsula cover 3 800 km$^2$ including the South Shetland islands where modelled MAGT was slightly below 0°C and the mountains of the south-eastern Antarctic Peninsula with modelled MAGT of around -19 °C. The modelled near-surface permafrost temperatures in the Antarctic Peninsula were the warmest among all Antarctic regions with an average modelled MAGT of -7.3 °C (Fig. 12).





Figure 12: Permafrost temperature maps of the Antarctic Peninsula and differences between borehole and modelled MAGT. See Fig. 1 for location of panels (a) and (b).

## 3.9.1 Palmer Land

The mountains of Palmer Land show considerable differences between the eastern and western parts of the peninsula. Modelled MAGTs at the mountains of the Orville Coast were around -17 °C, increasing to about -12 to -15 °C at the Black Coast and eventually rising above -10 °C at the Wilkins Coast. On the west side at the Fallieres Coast MAGTs of up to -4 °C were modelled. Temperatures decreased to -6 to -8 °C at the Rymill Coast, around -10 °C at 1000 m a.s.l., and approached -12°C at 1500 m a.s.l.. Similar MAGT ranges were modelled on Alexander Island where MAGTs close to the coast were around -5 °C, decreasing to between -9 and -7 °C at 1000 m a.s.l. and falling below -10 °C at elevations above 2000 m a.s.l.





### 3.9.2 Graham Land

A considerable increase in modelled MAGT from the east to the west of Graham Land was observed, where the south part of the east coast is protected by the Larsen Ice Shelf. Ground temperatures gradually increased in the northward direction on both sides of the peninsula. MAGTs at the Bowman and Foyn Coasts ranged between -8 and -6 °C and slowly increased from -6

5   °C at the Oscar II Coast to around -4 °C at the northern-most part of the mainland although still falling below -8 °C at higher elevations. On the west side, MAGT gradually decreased from around -2 °C in the north Davis Coast to -5 °C at the south of the Danco Coast and Anvers Island, where MAGT also fell below -6°C at higher elevations. Similar MAGT ranges were observed at the Graham and Loubet Coasts and at Adelaide Island but were lower than -7 °C at higher elevations. Small Islands along the west coast of the Antarctic Peninsula had a modelled MAGT of between -3 and -1 °C.

### 3.9.3 South Shetland Islands and James Ross Island

Another frequently studied area in the Antarctic is the Northern Antarctic Peninsula, especially the South Shetland Islands. At sites close to sea level, the modelled MAGT was usually above -1 °C on the South Shetland Islands (Fig. 13). MAGT decreased below -2 °C on sites that are not adjacent to the coast but still low-lying such as Byers and Hurd Peninsulas on Livingston

15   Island and Fildes Peninsula on King George Island. The MAGT was below -4 °C on the highest unglaciated peaks of Livingston and Smith Islands and reached -3 °C on Deception Island.

Modelled MAGTs were significantly colder on James Ross Island than on the South Shetland Islands, and, at sites adjacent to the sea, ranging from -3 °C in the north down to -5 °C in the south. Lower lying ice-free areas, including Seymour Island, had

20   MAGTs typically between -6 and -5 °C. The MAGTs on the highest rock outcrops were modelled as down to -7 °C.





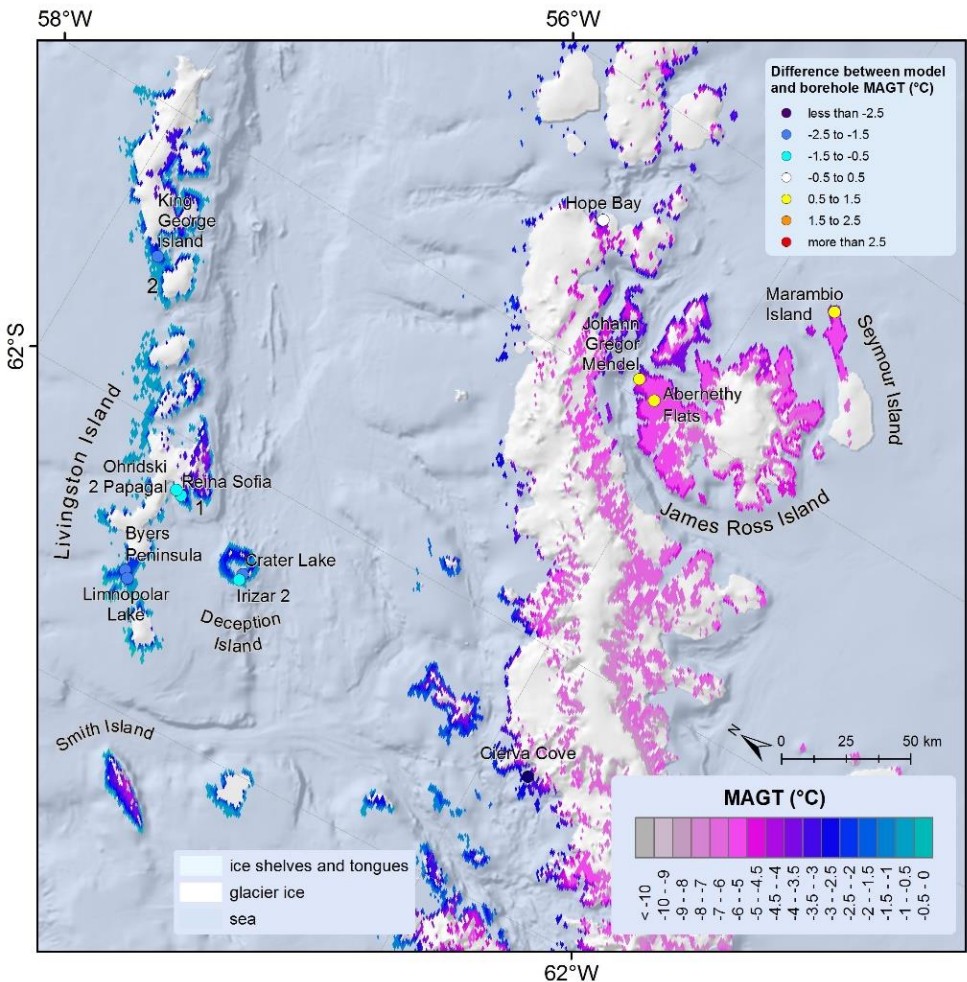

**Figure 13: Permafrost temperature map of the Northern Antarctic Peninsula and differences between borehole and modelled MAGT. See Fig. 1 for location. Note: the MAGT colour ramp is different from other figures. 1: Hurd Peninsula; 2: Fildes Peninsula.**

### 3.10 Other Antarctic and sub-Antarctic Islands

5   The modelling results were derived for the Antarctic Islands and sub-Antarctic Islands where the size and ice-free area of the island is sufficient that MODIS LST data were available. MAGT on Signy Island ranged from -1.5 °C at the coast to -4 °C in the interior. Permafrost was modelled on all South Sandwich Islands with similar MAGT ranges to those on Signy Island (Fig. 14). Less permafrost is modelled on South Georgia Island, where MAGTs at the coast increase to 1 °C and sporadic permafrost starts to occur at elevations above 100 m. The MAGTs decreased below -2 °C at the highest-lying rock outcrops. No permafrost

10  was modelled on the Crozet Islands. MAGTs below 0 °C were modelled at the highest elevations of Kerguelen Island and in isolated permafrost patches occurring above 500 m a.s.l. Permafrost is present also on the southern-most ice-free area of Heard Island.



**Figure 14: Permafrost temperature maps of Antarctic Islands and difference between borehole and modelled MAGT. See Fig. 1 for location of panels (a), (b), (c), (d) and (e). Note: the MAGT colour ramp is different from other figures.**



### 3.11 Regional MAGT distribution

The MAGT distribution of Antarctic is bimodal with the most pronounced peak at – 21 °C and the second peak at -7 °C (Fig. 15). The peaks correspond to the two largest ice-free areas of the Transantarctic Mountains and the Antarctic Peninsula, however the temperatures around the -20 °C peak also occur in other regions such as Queen Maud Land, the Vestfold Hills and the Ellsworth Mountains. The most commonly modelled temperatures were between -23 and -18 °C and usually occurred in the mountains rising above the Antarctic Ice-sheet and glaciers. MAGTs of between -10 and -6 °C occurred in the coastal areas of Wilkes Land, Marie Byrd Land, Queen Maud Land, Enderby Land and the Vestfold Hills, but the peak of temperature distribution at coastal sites shifted towards -7 °C because of MAGTs on the Antarctic Peninsula. The peak of coastal areas would be at -9 °C if the Antarctic Peninsula was excluded.

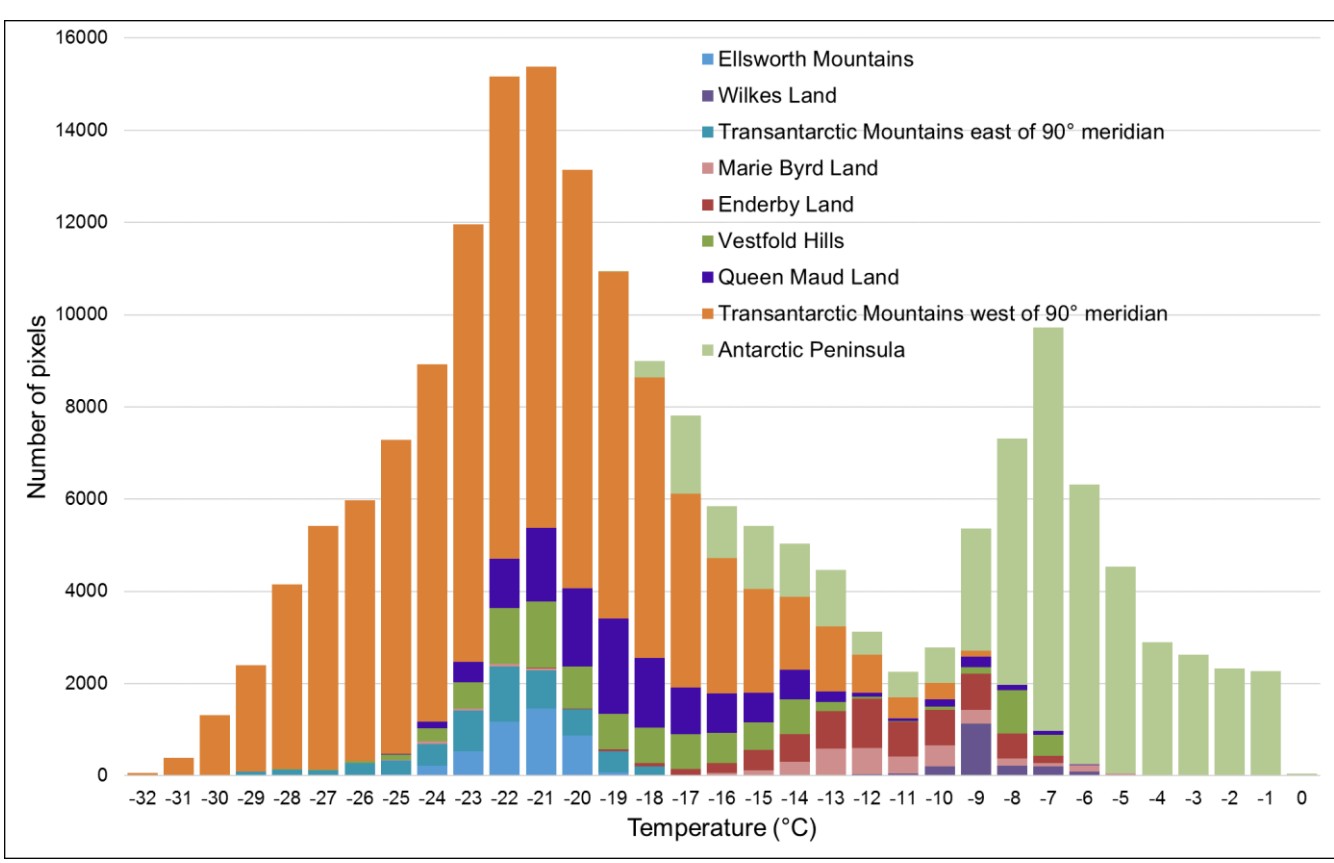

**Figure 15: Histogram of temperature distribution across the Antarctic for 1 °C bins. Note: The ice-free areas were often smaller than the analysed pixel size.**





### 3.12 Altitudinal MAGT gradients

Average MAGT lapse rate for the whole Antarctic was 0.40 °C 100 m$^{-1}$ ranging from 0.15 °C 100 m$^{-1}$ in the Ellsworth Mountains to 0.59 °C 100 m$^{-1}$ in Enderby Land. The lapse rates increased from 0.21 °C 100 m$^{-1}$ in Wilkes Land to 0.38 °C 100 m$^{-1}$ in the Transantarctic Mountains, 0.44 °C 100 m$^{-1}$ in the Vestfold Hills, 0.47 °C 100 m$^{-1}$ in Marie Byrd Land and in

the Antarctic Peninsula to 0.49 °C 100 m$^{-1}$ in Queen Maud Land.

The lapse rates are is indicating significant regional differences in the modelled MAGT patterns in relation to elevation (Fig. 16). The warmest among all regions was the Antarctic Peninsula which had MAGTs similar to Marie Byrd Land at around -12 °C between 1300 m a.s.l., which is again colder at higher elevations. Marie Byrd Land, Queen Maud Land, Enderby Land

and Wilkes Land had MAGTs of -9 °C at the coast but show varying characteristics of temperature decrease with elevation. Marie Byrd Land was the warmest with a decrease in MAGT up to 1500 m a.s.l., and a faster decrease to -25 °C at 3000 m a.s.l. The slower decrease in temperature with altitude to the 1500m elevation was also modelled in Enderby Land but the MAGT was colder. The MAGT decreased rapidly to -13 °C at 700 m a.s.l. in Queen Maud Land, where it slightly increased with elevation and then dropped steadily to -22 °C at 2800 m a.s.l.

The modelled MAGT in the Vestfold Hills dropped rapidly from -10 °C at the coast to -17 °C at 200 m a. s. l. and then gradually decreased to -24 °C at 2000 m a.s.l. There are no rock outcrops close to sea level in the Ellsworth Mountains and the MAGT at 100 m a.s.l. was -20 °C. In the Ellsworth Mountains there was a slow decrease of the modelled MAGT, to -25 °C at 3000 m a. s. l. The coldest MAGTs (below -30 °C) were modelled in the Transantarctic Mountains. In the part west of the 90 ° meridian,

the MAGT dropped from -20 °C at 500 m a.s.l to -27 °C at 2000 m a.s.l. but the absolute temperatures were lower east of the 90 ° meridian where elevations exceed 4000 m a.s.l.







**Figure 16: Altitudinal MAGT gradients for Antarctic Soil Regions calculated for 100 m elevation bins.**

According to the gradients for sub-regions of the Northern Antarctic Peninsula, the warmest were the South Shetland Islands

5     followed by Palmer Archipelago (Fig. 17). The altitudinal MAGT profiles showed clear differences between the western and eastern parts of the Northern Antarctic Peninsula mainland, although James Ross Island had a similar altitudinal profile to the West Antarctic Peninsula mainland. A significant decrease in MAGT from sea level to 100 m elevation was observed on the South Shetland Islands, Palmer Archipelago and the West Antarctic Peninsula mainland.

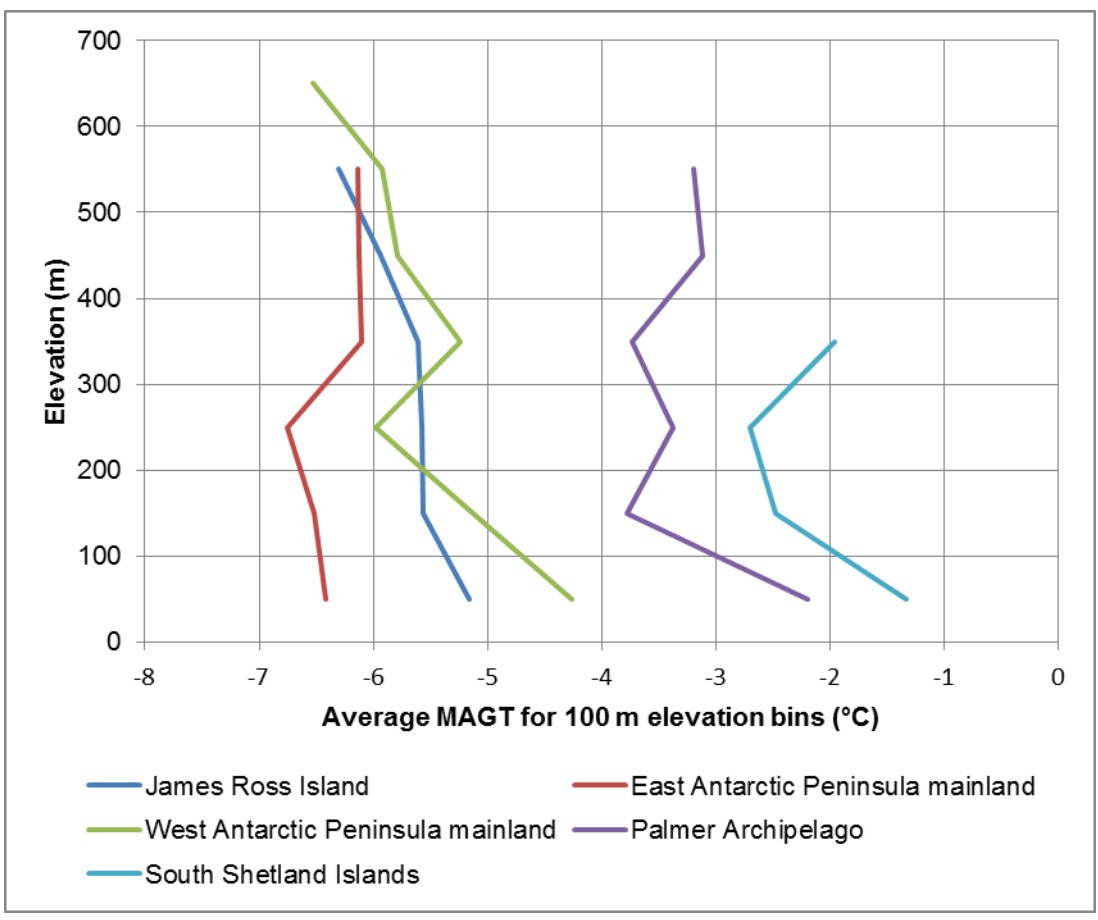

**Figure 17: Altitudinal MAGT gradients for the Northern Antarctic Peninsula calculated for 100 m elevation bins.**

## 4 Discussion

5   ### 4.1 Comparison to borehole measurements

### 4.1.1 Queen Maud Land

The modelled MAGT is according to validation data, was over-estimated in Queen Maud Land. Over-estimation in
Schirmacher Hills and Troll Station was below 1 °C but was higher than 4 °C in Flarjuven Bluff and Vesleskarvet. Both
stations are located on nunataks, which are exposed to wind that blows away the snow. The estimated 120 mm of annual
10   snowfall resulted in too low $n_f$-factors for snow-free conditions. However, the minimum MAGT of the ensemble spread
approaches the measured MAGT at both stations (Appendix A).





### 4.1.2 Enderby Land, Vestfold Hills and Wilkes Land

The borehole data in Enderby Land were from the Molodejnaya station (Thala Hills), where modelled MAGT was over-estimated by 1.6 °C. The over-estimation can be explained by rather thin snow cover due to strong winds and snow redistribution at the borehole site, which is confirmed by small differences between measured winter air and ground surface
temperatures. The MAGT is accurately modelled in the coastal parts of the Vestfold Hills region, where validation data are available. The difference between modelled and measured MAGT is small at the Larsemann Hills borehole but slight MAGT over-estimation (between 0.6 and 0.9 °C) was observed in comparison to the Larsemann and Landing Nunatak borehole measurements. Similarly, the MAGT was accurately modelled at the Bunger Hills station in the Wilkes Land region, with only small differences in comparison to measured ground temperature.

### 4.1.3 Transantarctic Mountains

The majority of the validation data in the Transantarctic Mountains region is available in the vicinity of the McMurdo Dry Valleys. The sites on the floors of the McMurdo Dry Valleys (Victoria Valley, Wright Valley Floor and Bull Pass) were modelled well with slight over-estimation in Victoria Valley and under-estimation of around 1 °C in the Wright Valley. Two
observation sites on small terraces on the walls of the Wright Valley (WV south wall and WV north wall) were modelled 3–4 °C too cold. The difference can be explained by the micro-location of the both sites, which are sheltered from katabatic winds, are above the winter inversion layer, and receive abundant summer solar radiation. The MAGT was over-estimated by up to 1 °C at Mt. Fleming, Scott Base and the Marble Point Borehole, which lie outside the valleys. The MAGT at the Marble Point site, characterised by glacial till, was over-estimated by 2.2 °C, however, the Marble Point Borehole, which was drilled in
granite bedrock approx. 1 km away, showed smaller over-estimation. The MAGT was under-estimated by more than 4 °C at the Granite Harbour site which has a warm microclimate as it is situated on north-facing moraine that receives a lot of meltwater from up-slope.

Outside the McMurdo Dry Valleys, MAGT measurements from the Zucchelli Station and Baker Rocks are available. The
MAGT is over-estimated by 2.3 °C at the Boulder clay borehole, which was drilled in a glacial till, exposed to katabatic winds, and characterised by numerous snow drifts (Guglielmin, 2006) which causes high local variability in permafrost conditions. On the other hand, the MAGT was over-estimated by only 0.2 °C at the Oasi New borehole, which is located in a granitic outcrop. MAGT at the Baker Rocks site, which is situated in littoral deposits (Raffi and Stenni, 2011), was under-estimated by 0.9 °C.

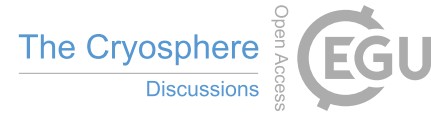

### 4.1.4 Marie Byrd Land

The only available borehole data in Marie Byrd Land is from the Russian research station Russkaya (the name of the borehole is Hobbs coast). According to the measured MAGT between 2008–2013 the modelled MAGT is over-estimated by 3.4 °C. The area is characterised by frequent storms and very strong winds, which blow off most of the snow on one hand and decrease

the number and quality of the MODIS measurements on the other hand. The snow free-conditions at the Hobbs coast borehole site were likely not simulated by the snow model, which resulted in the MAGT over-estimation. An alternative explanation could be the measurement period, which is only five years in comparison to 17 modelled years. No validation data were available for the higher elevations of the volcanic ranges. The modelled MAGTs there might be under-estimated because the TTOP model doesn't account for the ground heat flux. This can especially be the case in the locations with recent volcanic

activity.

### 4.1.5 Antarctic Peninsula

Comparison of the modelled MAGT with measured MAGT in boreholes showed an under-estimation in the Western Antarctic Peninsula and slight over-estimation in the Eastern Antarctic Peninsula. The MAGTs are under-estimated by between 1 and

2.1 °C on the South Shetland Islands (King George, Livingston and Deception Island). These under-estimations may be explained by heat advection from meltwater and rain that is not simulated by the model but is especially common in this part of the Antarctic. Another possible explanation for deviations of the modelled MAGT in the north-eastern Antarctic Peninsula are the frequent cloudy conditions. Although clouds are generally masked out from the MODIS LST and replaced by ERA Reanalysis temperatures, measurements in some areas are still contaminated with cloud temperatures, which results in MAGT

under-estimation (Østby et al., 2014).

Recent shallowing of thaw depth and ground cooling were observed on Deception Island by Ramos et al. (2017). However, similar cooling was recorded also in the Eastern Antarctic Peninsula, but MAGT was over-estimated by 1.4 °C at Marambio Island and by 0.8 °C at Abernethy Flats and 0.6 °C at the Johann Gregor Mendel borehole sites. MAGT was over-estimated

by only 0.2 °C at the Hope Bay mainland site which suggests that there was a continuous gradient of MAGT over-estimation from the Eastern Antarctic Peninsula to MAGT under-estimation on the Western Antarctic Peninsula.

### 4.2 Permafrost controls

The modelled permafrost temperatures reflect the climatic characteristics of the Antarctic with major controls of latitude,

elevation and continentality (Vieira et al., 2010). The effects of the ocean and continentality are well reflected in altitudinal MAGT profiles in Figs 16 and 17. Regions with areas close to the open sea generally show faster MAGT decrease with

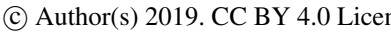

elevation, which was observed especially in the Vestfold Hills, the Transantarctic Mountains east of 90° meridian and in Marie Byrd Land where the MAGT dropped significantly with the first elevation increase of 100 or 200 m. The same phenomenon was observed in the North Antarctic Peninsula on the South Shetland Islands, Palmer Archipelago and on the West Antarctic Peninsula mainland where there is less sea ice presence in comparison to James Ross Island and the West Antarctic Peninsula mainland where this MAGT decrease was not observed. The continental mountainous regions of the Ellsworth Mountains and the Transantarctic Mountains west of the 90° meridian have, on the other hand, no open sea in the vicinity and had a significantly slower MAGT drop with the altitude.

The modelled MAGT lapse rate of 0.40 °C 100 m$^{-1}$ for the whole Antarctic is lower than the average air temperature lapse rate of 0.65 °C 100 m$^{-1}$ in the International Standard Atmosphere (ISO 2533:1975) but is, however, the same as the mean air temperature lapse rate measured for ice-free sites on James Ross Island from 2013–2016 (Ambrozova et al., 2019). Small increases of MAGTs with elevation were observed in many regions. This might be attributed to the nature of the analysis rather than a presence of temperature inversions. The rock outcrops on the scale of the regions are often present at different elevations far from each other, which results in occurrence of higher MAGTs at higher elevations. Additionally, some elevation bins have only a few rock outcrop pixels, which makes the averaged MAGT less representative for the whole region.

The lowest MAGT of -33.5 °C was modelled in the Transantarctic Mountains at the Queen Elizabeth Range, where the highest peak reaches 4350 m. The MAGT there could fall below -36 °C according to the coldest ensemble member. This is the lowest MAGT modelled on Earth according to the modelling in the other parts of the globe (Obu et al., 2019). On the highest Antarctic peak Mount Vinson in the Ellsworth Mountains reaching 4892 m a MAGT of only -26.1 °C was modelled. The Queen Elizabeth Range lies approximately 5° further south than Mount Vinson, illustrating the effect of latitude on permafrost temperatures.

The winter air temperature inversions that occur in the McMurdo Dry Valleys result in air temperatures that are approx. 10 °C lower in the valleys than in the surroundings, but inversions are occasionally disrupted by katabatic winds from the polar ice sheet (Nylen et al., 2004). The winter inversions reflect in MAGTs being approx. 3 °C colder on the floor than in surroundings in Victoria Valley, where the inversions are particularly intense in comparison to other valleys. No MAGT inversions were modelled in the Wright Valley, which lies only 50 m a.s.l (though winter temperature inversions do occur there), or in the Taylor Valley, which is opened to the coast and can drain cold air.

### 4.3 Model performance and limitations

As the LST data are primarily derived from satellite data, their availability and accuracy depend on cloud cover. On one hand the frequent cloud cover might be a reason for general under-estimation of modelled MAGT at Antarctic Peninsula. On the

other hand the clear sky conditions in the McMurdo Dry Valleys might explain the relatively successful modelling results in this area.

The absence of vegetation in the Antarctic results in high snow redistribution by wind and highly spatially variable snow cover, which influences ground temperatures in many parts of Antarctic (Guglielmin et al., 2014; Ramos et al., 2017; Ferreira et al. 2017). Neither snow redistribution nor sublimation are simulated by our snowfall model and the average snow depths could not be estimated and $n_f$-factors derived as for the Northern Hemisphere by Obu et al. (2019). Although an ensemble of 200 model runs with varying annual snowfall is used, the mean of the ensemble runs does not always represent the borehole site microclimate and ground properties. In the case of wind-exposed nunataks where the snow presence is over-estimated on larger areas, such as Flarjuven Bluff and Vesleskarvet, the MAGTs can be over-estimated up to 4 °C, but the modelled ensemble minimum still approaches the measured MAGT.

Several permafrost and active-layer studies in the Antarctic have noted occurrence of $n_f$-factors above 1 (Lacelle et al., 2016; Kotzé and Meiklejohn, 2017), which indicates that average air temperatures are higher than ground surface temperatures. The likely explanation could be a presence of snow during the warmer part of the year, which is insulating ground from heat, unlike from cold in the winter. However Kotzé and Meiklejohn (2017) mention also a presence of blocky deposits at the Vesleskarvet site, which could result in ground cooling due to cold air advection. The concept of $n_f$-factors was introduced for the Northern Hemisphere for snowcover effect during freezing conditions (Smith and Riseborough, 1996), which challenges the derivation of $n_f$-factors for TTOP modelling on sites with highly temporarily variable snow cover.

Elevation is one of the major permafrost controlling factors in the Antarctic (Vieira et al., 2010). In steep terrain, the model input datasets are less likely to be representative for the micro-locations present within the modelled pixel or the borehole site as for example shown on the boreholes sites on the walls of the Wright Valley. A number of the Antarctic boreholes are situated in mountainous environments, which might explain some of the discrepancies between measured and modelled MAGT. Elevation uncertainties in DEM are inherited by the model and reflected in the estimated average annual snowfall and downscaled ERA reanalysis temperatures. The elevations of some peaks might also not be well represented at the spatial resolution of 1 km, therefore the modelled MAGT might appear warmer than MAGT found on the top a peak.

## 5 Conclusions

Near-surface permafrost temperatures in the Antarctic were most commonly modelled as between -23 and -18 °C for mountainous areas rising above the Antarctic Ice Sheet. The Earth's lowest permafrost temperature of - 33 °C was modelled at Mount Markham in Queen Elizabeth Range in the Transantarctic Mountains. Coastal regions were usually characterised with ground temperatures of between -14 and -8 °C, approaching 0 °C in the coastal areas of the Antarctic Peninsula and rising

above 0°C in the Antarctic Islands. The regional variations in permafrost temperatures can be explained by (1) continentality, which influences permafrost temperatures especially, at elevations of up to 200 m; (2) elevation; and (3) latitude, which explains differences in permafrost temperatures at similar elevations.

Comparison of modelled temperatures to 40 permafrost boreholes and soil climate stations yielded root mean square error of 1.9 °C but the accuracy varied significantly among borehole sites. The difference was smaller than 1 °C for more than 50 % of the sites, but can exceed 4 °C. The greatest differences between the modelled and measured permafrost temperatures occurred where snow conditions were not successfully represented in the model. These sites are generally exposed to a strong wind-driven redistribution of snow as for example at nunataks in Queen Maud Land, on the Hobs Coast, and in Marie Byrd

Land. Considerable differences between modelled and measured MAGTs also occurred at sites with microclimate and ground properties that are not representative for the respective modelled 1 km$^2$ pixel. Permafrost temperatures on the walls of Wright Valley and in Granite Harbour were under-estimated by up to 4 °C, which can be explained by warm microclimates of the borehole sites compared to surroundings. The model performed well in areas with frequent cloud-free conditions such as the McMurdo Dry Valleys, where even winter air temperature inversions are reflected in the modelled permafrost temperatures.

Frequent cloudy conditions in the north-western Antarctic Peninsula can to some extent explain the systematic under-estimation of modelled permafrost temperatures in this area.

This study is the first continent-wide modelling of permafrost temperatures for the Antarctic. It reports near-surface permafrost temperatures for remote regions without observations, which is highly valuable for research fields, such as climate change,

terrestrial ecology, microbiology or astrobiology. Our study suggests that extended networks of currently sparse borehole temperature measurements and spatially distributed information on snow cover and ground properties are crucial for improving future permafrost modelling results in the Antarctic.

**Data availability:** The data are available for download at: https://doi.pangaea.de/10.1594/PANGAEA.902576





**Appendix A: List of borehole properties, measurements and modelled results**

| Borehole name | Lattitude | Longitude | Sensor depth (cm) | Elevation (m) | Measured MAGT (°C) | MAGT calculation period | Modelled mean MAGT (°C) |
|---|---|---|---|---|---|---|---|
| Johann Gregor Mendel | -63.80000 | -57.86670 | 75 | 10 | -5.60 | 2011-2017 | -4.98 |
| Abernethy Flats | -63.88140 | -57.94830 | 75 | n/a | -6.15 | 2006-2016 | -5.38 |
| Bunger Hills | -66.27530 | 100.76000 | 500 | 7 | -8.90 | 2008-2014 | -9.09 |
| Schirmacher Hills | -70.77177 | 11.73673 | 100 | 80 | -8.50 | 2009-2016 | -8.19 |
| Larsemann Hills | -69.38669 | 76.37538 | 500 | 96 | -7.80 | 2013-2015 | -7.83 |
| Larsemann | -69.40421 | 76.34465 | 300 | 96 | -8.60 | 2008, 2010-2015 | -7.90 |
| Landing nunatak | -69.74781 | 73.70503 | 100 | 96 | -11.00 | 2011-2012 | -10.08 |
| King George island | -62.19667 | -58.96556 | 500 | 20 | -0.70 | 2008-2009, 2014 | -2.21 |
| Hobs coast | -74.76333 | -136.79639 | 50 | 76 | -10.30 | 2008-2013 | -6.86 |
| Molodejnaya | -67.66556 | 45.84194 | 50 | 45 | -9.40 | 2008, 2011-2013, 2015-2016 | -7.81 |
| Reina Sofia | -62.67028 | -60.38222 | n/a | 275 | -1.78 | n/a | -3.05 |
| Cierva Cove | -64.16195 | -60.95093 | 1500 | 182 | -0.95 | n/a | -3.67 |
| Amsler | -64.77619 | -64.06057 | 900 | 67 | -0.36 | 2016-2017 | -1.48 |
| Crater Lake | -62.98333 | -60.66667 | n/a | 85 | -0.83 | n/a | -2.96 |
| Byers Peninsula | -62.62981 | -61.06013 | n/a | 92 | -0.43 | n/a | -2.33 |
| Limnopolar Lake | -62.64959 | -61.10405 | 130 | 90 | -0.60 | 2009-2012 | -2.34 |
| Rothera Point | -67.57070 | -68.11879 | n/a | 31 | -3.10 | 2009-2011 | -2.78 |
| Marambio Island | -64.23333 | -56.61667 | n/a | 200 | -6.60 | 2009-2012 | -5.24 |
| Signy Island | -60.71655 | -45.59978 | n/a | 90 | -2.10 | 2006-2009 | -2.11 |
| Ohridski 2 Papagal | -62.64811 | -60.36375 | 400 | 147 | -1.04 | 2008-2018 | -1.90 |
| Irizar 2 | -62.98263 | -60.71562 | 80 | 130 | -1.58 | 2009-2017 | -2.49 |
| Troll Station | -72.01139 | 2.53306 | 3 | 1275 | -17.40 | 2007–2015 | -17.09 |
| Flarjuven Bluff | -72.01167 | -3.38833 | 3 | 1220 | -17.50 | 2008–2015 | -13.06 |
| Vesleskarvet | -71.68998 | -2.84758 | 3 | 805 | -16.10 | 2009–2014 | -11.91 |
| Boulder Clay | -74.74583 | 164.02139 | n/a | 205 | -16.90 | 1996-2009 | -14.63 |
| Oasi New | -74.70000 | 164.10000 | n/a | 80 | -13.50 | 2005-2009 | -13.32 |
| Bull Pass | -77.51847 | 161.86269 | 60 | 141 | -19.44 | 2000-2017 | -20.93 |
| WV south wall (Bull Pass East) | -77.50219 | 162.06475 | 50 | 832 | -16.44 | 2013-2017 | -20.04 |
| WV north wall (Don Juan Pond) | -77.57388 | 161.23877 | 50 | 734 | -16.76 | 2011-2017 | -20.83 |
| Granite Harbour | -77.00655 | 162.52561 | 66 | 6 | -14.31 | 2003-2017 | -18.53 |
| Marble Point | -77.41955 | 163.68247 | 120 | 47 | -18.15 | 2000-2017 | -15.99 |
| Minna Bluff | -78.52500 | 166.78240 | 43 | 32 | -16.44 | 2007-2016 | -16.52 |
| Mt. Fleming | -77.54519 | 160.29027 | 22 | 1697 | -23.81 | 2002-2017 | -23.58 |
| Scott Base | -77.84831 | 166.76058 | 40 | 44 | -17.42 | 2000-2017 | -16.64 |
| Victoria Valley | -77.33178 | 161.60069 | 30 | 410 | -22.87 | 2000-2017 | -22.97 |
| Wright Valley Floor | -77.51808 | 161.85117 | 75 | n/a | -19.13 | 2000-2017 | -20.93 |
| Marble Point Borehole | -77.40732 | 163.72913 | 200 | 85 | -16.90 | 2009-2015 | -15.96 |
| Baker Rocks | -74.20750 | 164.83361 | 3 | 11 | -15.60 | 2006–2015 | -16.52 |
| Mt. Dolence | -79.82181 | -83.19714 | 30 | 886 | -18.30 | 2012-2013 | -20.21 |
| Hope Bay | -63.40635 | -56.99656 | 80 | n/a | -4.10 | 2010 | -3.83 |



## Appendix A: (Continuation)

| Borehole name | Difference between modelled and measured MAGT (°C) | Modelled max MAGT | Modelled min MAGT | Modelled SD (°C) | Source | Soil Region |
|---|---|---|---|---|---|---|
| Johann Gregor Mendel | 0.62 | -2.83 | -6.43 | 0.89 | Hrbáček et al., 2017a | Antarctic Peninsula |
| Abernethy Flats | 0.77 | -2.89 | -6.89 | 1.08 | Hrbáček et al., 2017b | Antarctic Peninsula |
| Bunger Hills | -0.19 | -4.41 | -11.50 | 1.41 | Andrey Abramov | Wilkes Land |
| Schirmacher Hills | 0.31 | -3.91 | -9.95 | 1.08 | Andrey Abramov | Queen Maud Land |
| Larsemann Hills | -0.03 | -3.53 | -9.82 | 1.23 | Andrey Abramov | Vestfold Hills |
| Larsemann | 0.70 | -3.64 | -9.76 | 1.17 | Andrey Abramov | Vestfold Hills |
| Landing nunatak | 0.93 | -6.10 | -12.39 | 1.33 | Andrey Abramov | Vestfold Hills |
| King George island | -1.51 | -0.93 | -2.75 | 0.47 | Andrey Abramov | Antarctic Peninsula |
| Hobs coast | 3.44 | -4.06 | -10.14 | 1.66 | Andrey Abramov | Marie Byrd Land |
| Molodejnaya | 1.59 | -4.06 | -10.19 | 1.66 | Andrey Abramov | Enderby Land |
| Reina Sofia | -1.27 | -1.54 | -3.97 | 0.67 | Miguel Ramos | Antarctic Peninsula |
| Cierva Cove | -2.72 | -1.95 | -5.51 | 0.86 | Gonçalo Vieira | Antarctic Peninsula |
| Amsler | -1.12 | -0.65 | -2.42 | 0.49 | Gonçalo Vieira | Antarctic Peninsula |
| Crater Lake | -2.13 | -1.51 | -4.01 | 0.65 | Gonçalo Vieira | Antarctic Peninsula |
| Byers Peninsula | -1.90 | -1.10 | -2.95 | 0.50 | Oliva et al. (2017) | Antarctic Peninsula |
| Limnopolar Lake | -1.74 | -1.04 | -2.90 | 0.47 | de Pablo et al. (2014) | Antarctic Peninsula |
| Rothera Point | 0.32 | -1.36 | -3.95 | 0.74 | Guglielmin et al. (2014) | Antarctic Peninsula |
| Marambio Island | 1.36 | -3.01 | -6.58 | 0.83 | Jorge Strelin | Antarctic Peninsula |
| Signy Island | -0.01 | -0.97 | -2.86 | 0.45 | Guglielmin et al. (2012) | Antarctic Islands |
| Ohridski 2 Papagal | -0.87 | -0.93 | -2.80 | 0.43 | Gonçalo Vieira | Antarctic Peninsula |
| Irizar 2 | -0.91 | -1.21 | -3.33 | 0.60 | Gonçalo Vieira | Antarctic Peninsula |
| Troll Station | 0.31 | -9.41 | -19.56 | 1.19 | Hrbáček et al. (2018) | Queen Maud Land |
| Flarjuven Bluff | 4.45 | -6.46 | -16.84 | 2.68 | Hrbáček et al. (2018) | Queen Maud Land |
| Vesleskarvet | 4.19 | -6.01 | -15.09 | 2.57 | Hrbáček et al. (2018) | Queen Maud Land |
| Boulder Clay | 2.27 | -8.83 | -16.78 | 1.22 | Vieira et al. (2010) | Transantarctic Mountains |
| Oasi New | 0.18 | -8.24 | -15.08 | 0.84 | Vieira et al. (2010) | Transantarctic Mountains |
| Bull Pass | -1.49 | -17.24 | -22.65 | 0.95 | USDA (Seybold et al., 2009) | Transantarctic Mountains |
| WV south wall (Bull Pass East) | -3.60 | -17.35 | -21.64 | 0.90 | Megan Balks | Transantarctic Mountains |
| WV north wall (Don Juan Pond) | -4.07 | -18.33 | -22.50 | 0.82 | Megan Balks | Transantarctic Mountains |
| Granite Harbour | -4.22 | -15.52 | -20.15 | 0.84 | USDA (Seybold et al., 2009) | Transantarctic Mountains |
| Marble Point | 2.16 | -13.67 | -17.83 | 0.65 | Megan Balks | Transantarctic Mountains |
| Minna Bluff | -0.07 | -10.74 | -19.03 | 1.35 | Megan Balks | Transantarctic Mountains |
| Mt. Fleming | 0.23 | -19.36 | -25.66 | 1.01 | Megan Balks | Transantarctic Mountains |
| Scott Base | 0.78 | -10.72 | -18.84 | 1.08 | Megan Balks | Transantarctic Mountains |
| Victoria Valley | -0.10 | -21.26 | -24.59 | 0.90 | Megan Balks | Transantarctic Mountains |
| Wright Valley Floor | -1.79 | -17.24 | -22.65 | 0.95 | Megan Balks | Transantarctic Mountains |
| Marble Point Borehole | 0.95 | -11.46 | -17.72 | 0.83 | Guglielmin et al. (2011) | Transantarctic Mountains |
| Baker Rocks | -0.92 | -7.11 | -19.39 | 1.59 | Hrbáček et al. (2018) | Transantarctic Mountains |
| Mt. Dolence | -1.91 | -13.48 | -22.37 | 1.13 | Schaefer et al. (2017b) | Ellsworth Mountains |
| Hope Bay | 0.27 | -1.82 | -4.97 | 0.66 | Schaefer et al. (2017a) | Antarctic Peninsula |





**Author contribution:** JO, SW, GV, AB and AK designed conceptual framework for the study and the model was developed and ran by SW and JO. Ground validation data was contributed by GV, AA, MB, FH and MR, who also provided interpretation of results on regional scale. JO wrote the paper based from input and feedback from all co-authors.

**Competing interests:** The authors declare that they have no conflict of interest.

**Acknowledgements:** This work was funded by the European Space Agency Data User Element GlobPermafrostt project in cooperation with ZAMG (grant number 4000116196/15/I-NB) and the Research Council of Norway SatPerm project (grant
number 239918). Data storage resources were provided by Norwegian National Infrastructure for Research Data (project NS9079K). The work of FH was supported by the Ministry of Education Youth and Sports of Czech Republic large infrastructure project LM2015078. Temperature data for Russian stations were obtained with the support form the Russian Antarctic Expedition and Government program AAAA-A18-118013190181-6. The PERMANTAR observatories in Western Antarctic Peninsula have been funded mainly by the Portuguese Foundation for Science and Technology and the Portuguese
Polar Program. The Terra and AQUA MODIS LST datasets were acquired from the Level-1 and Atmosphere Archive & Distribution System (LAADS) Distributed Active Archive Center (DAAC), located in the Goddard Space Flight Center in Greenbelt, Maryland (https://ladsweb.nascom.nasa.gov/). Over the years many people have contributed to installation and maintenance of the McMurdo Dry Valley Soil Climate stations, but particular thanks are due to Cathy Seybold, Ron Paetzold and Don Huffman from USDA, Jackie Aislabie and Fraser Morgan, Landcare Research, N.Z., and Chris Morcom, Dean
Sandwell, and Annette Carshalton, University of Waikato, NZ. Antarctica New Zealand provided logistic support for annual station access. Authors thank also to Nikita Demidov, Andrey Dolgikh, Elya Zazovskaya, Nikolay Osokin, Dima Fedorov-Davidov, Andrey Ivashchenko, Alexey Lupachev and Nikita Mergelov for borehole data retrieval from the Russian Antarctic Stations.




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
