# Peer review of "Pan-Antarctic map of near-surface permafrost temperatures at 1 km2 scale"

_The Cryosphere, 2019_

## Short Comment (SC1) · 25 Jun 2019

After a quick glance at the paper, I could provide some comments which are not ordered in any coherent way.

1. The mean absolute error in text is 0.06°C while on Fig.2 it is -0.017°C.

2. It is unclear what do authors mean by "values were set to randomly vary by $\pm0.1$ between 0.75 and 0.95" in section 2.5 with regards to $r_f$ parameter. Are values drawn from a uniform distribution between 0.75 and 0.95? What is the rationale behind the distribution of $r_f$?

3. In section 2.2, it is unclear if the downscaled ERA-interim and ERA-5 near-surface temperatures were debiased towards MODIS record to create a consistent record.
Since authors mention negative bias caused by data gaps wouldn't filling these gaps with near-surface temperature (which is not equivalent to surface temperature in general) provide additional bias to the record?

4. In the same section (2.2), it is unclear why LST data was averaged to 8 day periods? Since FDD and TDD are essentially numerical integrals it is unclear why 8-day binning would give any advantage.

---

## Referee Comment (RC1) · Joseph Levy (Referee) · 13 Aug 2019

Review of

Pan-Antarctic map of near-surface permafrost temperatures at 1 km$^2$ scale

tc-2019-148

Joseph Levy

**General Comments.**

This is an extremely interesting paper and represents a large step forward in efforts to understand the distribution of permafrost temperature in the Antarctic. It is a novel contribution and will likely be a benchmark set of predictions for years to come.

Much of the model work here relies on use of ERA-Interim reanalysis products and down-scaling. How do the reanalysis surface temperatures and precipitation values compare to measured surface temperatures and precipitation values at long-term monitoring sites? The model will only be as good as the measurements that underlie it, and microclimate conditions dominate many ice-free areas.

One potential shortcoming of the model is that all surfaces are treated as having the same composition in terms of thermal properties. Yet IFAs can be either solid bedrock (with a variety of different lithologies possible), or soil, or ice cemented soil. How much of the overall uncertainty in the model can be attributed to the choice of subsurface thermal properties? What about active layer characteristics? Where present, wet active layers may significantly increase heat transport into the surface, while dry active layers serve to insulate permafrost.

There are abundant examples of unclear English usage in this paper that should be addressed by a copy editor prior to submission. Many are small, e.g., "the temperature" when "temperature" would suffice, but they are too numerous for me to highlight individually.

**Specific Comments.**

As a courtesy to reviewers, please remember that continuous line numbers are always more convenient to reference.

P1, Line 20. "under" seems like a slightly odd word choice. Technically, it is mostly correct, since in most ice-free areas there is an active layer, although, not all ice-free areas have active layer conditions. For that reason, "within" might be a more accurate word to use here and throughout, unless subsurface temperatures are being discussed. See, P2, L2

P1, Line 21. "a" is not needed here

P2, L2-4. This sentence has several unclear uses of language and should be revised.

P2, L4. The Landsat-based reconstructions of ice-free area were conducted even earlier in the largest IFA by Levy (2012).

P5, L14. There is commonly not just an elevation gradient in Antarctica, but also a coastal-inland gradient for precipitation. How does proximity to coastal moisture sources affect snow cover at inland sites?

P6, L10. $r_k$ of 0.85 seems very high—especially for sandy, low-organic permafrost typical of the Transantarctics. (McKay et al., 1998) report thermal conductivities of thawed and ice-cemented permafrost of 0.6 and 2.5 W/mK, respectively, which would be less than 1/3 of the proposed value. (Levy and Schmidt, 2016) report a range of wetted soil thermal conductivity values for similar cold desert soils, ranging from ~0.2 to ~1.7 W/mK, which are still on the low end compared to the frozen value. What is the impact of having such a high $r_k$ value?

Fig. 3. It is a little bit confusing having different legends on the two maps, which show both datasets. Can you put both legends on both maps?

P13, L4. East of 90˚E would be a more clear way to phrase this.

Figs 3 – 14. I'd suggest thinking more about the color ramp used to present this temperature data. Why was it selected? Because it is not a single gradient or ramp, it is difficult to tell what color is warmer or cooler than which other color without consulting the legend. It is not extremely intuitive. Either a single color intensity ramp, or a red-blue or green-blue color ramp might be more clear.

P24, L10. Is part of the apparent lapse rate behavior that you have observed a function of changing substrate? In low elevation areas, tills and other soils may dominate, while at high elevations, ice-free areas are largely bedrock. How might this affect the apparent MAGT lapse rate?

P24, L14. The increase in MAGT with elevation at some sites is extremely curious. What do you attribute it to? Is this explained in P30, L25? It might be worth highlighting this curious result sooner.

P27, L4. Given the importance of orientation and insolation in controlling ground surface temperature in Antarctic ice free areas, would it be worth adding an orientation factor to the model? Something like fraction of north-facing surfaces in the grid cell? With datasets like REMA now available, it might significantly reduce the offsets in these numerous "microclimate" sites?

P31, L5. An RMSE of 1.9˚C is really pretty good, but it could be a substantial uncertainty for warmer permafrost sites. How does uncertainty scale with absolute or mean average temperature? Are there particularly warm sites that have high uncertainty? How might the distribution of uncertainty affect our understanding of which permafrost is at risk of thaw?

References.

Levy, J., 2012. How big are the McMurdo Dry Valleys? Estimating ice-free area using Landsat image data. Antarctic Science 25, 119–120. doi:10.1017/S0954102012000727

Levy, J.S., Schmidt, L., 2016. Thermal Properties of Antarctic Soils: Wetting Controls Subsurface Thermal State. Antarctic Science 28, 361–370.

McKay, C.P., Mellon, M.T., Friedmann, E.I., 1998. Soil temperatures and stability of ice-cemented ground in the McMurdo Dry Valleys, Antarctica. Antarctic Science 10, 31–38.

---

## Referee Comment (RC2) · Anonymous Referee #2 · 5 Oct 2019

Review for "Pan-Antarctic map of near-surface permafrost temperatures at 1km2 scale"

Authors have demonstrated a continent scale high resolution modeling work for Antarctic ground temperatures. The Cryogrid model is utilized for the ice-free areas of Antarctic continent and islands, as MODIS land surface temperatures together with downscaled ERA-Interim climate data is used for model forcing. Model results are compared to 40 borehole sites in different locations. Model performance is subject to subgrid heterogeneity, forcing data scarcity, and continentality/topography of the locations. Overall, the manuscript presents a first continent scale simulated ground temperature map for Antarctica that could be used as a guideline for different fields of science.

General comments:

[Figure]

1. Since the model relies heavily on input data such as satellite observation, it is best to include uncertainty ranges for areas where satellite observations are lacking (cloudy days) or low quality. Otherwise it is wrong to advertise these model results as a generally accepted guideline for Antarctic permafrost conditions.

2. Figures can be improved by proper color choice following other referee's suggestions. Also an overall figure of whole Antarctica with the modelled MAGT should be added at the end to give a general idea of MAGT in the whole continent to guide future studies. Forcing data uncertainty can be added to that figure.

3. The model limitations should also include possible sources of uncertainty for nf,nt, and rk values such as different thermal properties, soil hydrological conditions etc. Also it would be helpful to mention Cryogrid's difference to process-based models that are mainly used for northern hemisphere permafrost studies.

Other than these, I find this paper to be well worthy of the journal and Antarctic research, so I support this paper to be published with the minor improvements I listed.

Minor comments: - fig2: correct the mismatch MAE written on figure (-0.17) and in text (0.06) - stay consistent with abbreviations, MAGTs vs MAGTS (p8l16) - p24l7 correct "The lapse rates are is indicating..." - for section 4, please add corresponding figure numbers when discussing specific sites - conclusion at p31lines1-3, snow cover and redistribution effects should be added as an important factor as seen from the validations of this paper

---

## Author Comment (AC1) · 29 Oct 2019

Responses to the comments, revised manuscript and marked track changes are attached as tc-2019-148_revised.zip file.

Please also note the supplement to this comment:
https://www.the-cryosphere-discuss.net/tc-2019-148/tc-2019-148-AC1-supplement.zip

---

## Author Response (AR2)

Authors are grateful for constructive comments from all the referees that helped to improve the manuscript. A point-by-point response to the reviews and a list of all relevant changes can be found below. A marked-up manuscript version is added at the end of the document.

**Response to RC1: 'Comments on tc-2019-148',**

This is an extremely interesting paper and represents a large step forward in efforts to understand the distribution of permafrost temperature in the Antarctic. It is a novel contribution and will likely be a benchmark set of predictions for years to come.

**Comment:** Much of the model work here relies on use of ERA-Interim reanalysis products and downscaling. How do the reanalysis surface temperatures and precipitation values compare to measured surface temperatures and precipitation values at long-term monitoring sites? The model will only be as good as the measurements that underlie it, and microclimate conditions dominate many ice-free areas.

Response: The ERA reanalysis near-surface temperature data dominates over MODIS LST only in very cloudy area as the Western Antarctic Peninsula. The ERA reanalysis quality significantly affects results in these regions, whereas it has less influence in areas with little cloud cover. Jones and Lister (2015) compared near surface temperatures between meteorological stations and ERA-Interim reanalysis and reported that bias for the stations outside the ice sheet is - 0.9 °C with a RMSE of 1.96 °C, which is within the range of our model uncertainties. On the other hand, it is challenging to measure snowfall in the Antarctic, thus it is also difficult to estimate the accuracy of the snowfall used for the modelling.

**Comment:** One potential shortcoming of the model is that all surfaces are treated as having the same composition in terms of thermal properties. Yet IFAs can be either solid bedrock (with a variety of different lithologies possible), or soil, or ice cemented soil. How much of the overall uncertainty in the model can be attributed to the choice of subsurface thermal properties? What about active layer characteristics? Where present, wet active layers may significantly increase heat transport into the surface, while dry active layers serve to insulate permafrost.

Response: The authors would be very happy to include any information about ground properties for defining rk-factor, but such information is available only for very limited areas in the Antarctic and no suitable product is available for pan-Antarctic scale. The used TTOP model calculates only MAGTs without taking in to account any active layer properties. The only way that lithology influences the modelled results is through the rk-factor, which is a ratio between thawed and frozen thermal conductivity. It is used to scale TDDs thus it has affect only on results where TDDs are considerably high. The majority of the Antarctic regions have very little or no TDDs, thus the effect of using wrong rk-factors is negligible in most areas. A sensitivity test on the Antarctic Peninsula, where TDDs are the highest, showed that considerable changes in rk-factors had relatively small differences in the resulting MAGTs, which were in order of a few 0.1 °C.

Changes: A paragraph discussing potential shortcomings regarding the thermal properties of the ground and rk-factors was added to the section 4.3

**Comment:** There are abundant examples of unclear English usage in this paper that should be addressed by a copy editor prior to submission. Many are small, e.g., "the temperature" when "temperature" would suffice, but they are too numerous for me to highlight individually.

Response: The English language has been reviewed by a competent native English speaking technical writer and corrected as appropriate.

**Specific Comments.**

**Comment:** P1, Line 20. "under" seems like a slightly odd word choice. Technically, it is mostly correct, since in most ice-free areas there is an active layer, although, not all ice-free areas have active layer conditions. For that reason, "within" might be a more accurate word to use here and throughout, unless subsurface temperatures are being discussed. See, P2, L2

Changes: Changed accordingly to within

Comment:: P1, Line 21. "a" is not needed here

Changes: Changed accordingly

Comment: P2, L2-4. This sentence has several unclear uses of language and should be revised.

Response: The second part of the sentence was removed.

Changes: Sentence shortened to: "Permafrost in the Antarctic is present beneath all ice-free terrain, except for the lowest elevations of the maritime Antarctic and sub-Antarctic islands (Vieira et al., 2010)."

**Comment:** P2, L4. The Landsat-based reconstructions of ice-free area were conducted even earlier in the largest IFA by Levy (2012).

Response: We report only an ice-free area for the whole Antarctic in this part of the paper. The calculations from Burton-Johnson et al. (2016) were corrected by Hrbáček et al. (2018) and this is the reason that the both studies are cited.

Changes: The area was updated for McMurdo Dry Valleys in the section 3.6.1

**Comment:** P5, L14. There is commonly not just an elevation gradient in Antarctica, but also a coastal-inland gradient for precipitation. How does proximity to coastal moisture sources affect snow cover at inland sites?

Response: The snowfall was downscaled from Era-Interim and ERA-5 to 1 km2 spatial resolution. The coastal-inland gradients seem to be well represented as shown on the figure below.

**Comment:** P6, L10. rk of 0.85 seems very high—especially for sandy, low-organic permafrost typical of the Transantarctics. (McKay et al., 1998) report thermal conductivities of thawed and ice-cemented permafrost of 0.6 and 2.5 W/mK, respectively, which would be less than 1/3 of the proposed value. (Levy and Schmidt, 2016) report a range of wetted soil thermal conductivity values for similar cold desert soils, ranging from ~0.2 to ~1.7 W/mK, which are still on the low end compared to the frozen value. What is the impact of having such a high rk value?

Response: According to the input data, there are no thawing degree days in the Transantarctic region. TDDs at coastal sites contribute at most 0.2 % to the sum of TDDs and FDDs. Since only TDDs are multiplied by rk-factors in the TTOP model, the effect of rk factors is negligible in the Transantarctic region.

**Comment:** Fig. 3. It is a little bit confusing having different legends on the two maps, which show both datasets. Can you put both legends on both maps?

Response: Putting both legends on both panels would be feasible for this figure. Repeating this for all of the figures with two or more panels would mean significant increase in size of the figures, which already represent a significant part of the paper. The authors would therefore prefer just to state in the caption that legends are valid for all of the panels in the figure.

Changes: A sentence "Legends, scale, and north arrow are valid for both panels" was added to all figures with panels.

**Comment:** P13, L4. East of 90°E would be a more clear way to phrase this.

Changes: Changed accordingly.

**Comment:** Figs 3 - 14. I'd suggest thinking more about the color ramp used to present this temperature data. Why was it selected? Because it is not a single gradient or ramp, it is difficult to tell what color is warmer or cooler than which other color without consulting the legend. It is not extremely intuitive. Either a single color intensity ramp, or a red-blue or green-blue color ramp might be more clear.

**Response:**

The temperature ramp was created to represent the MAGTs for the Northern Hemisphere modelling with 21 classes that could not be distinguished using continuous colour ramp with only one or two colours and so many different hues. The same ramp was used also to represent MAGTs for the Antarctic. The warmer-coloured stretching from yellow to red was used for positive temperatures. Except for the Figure 14, there are no positive MAGTs to be displayed, so only the cold temperature part of the ramp, stretching from green, blue, purple to grey, was used for the most of the figures.

We believe that the chosen colour scheme is a good compromise between putting as much information as possible on the map and the ability for a scientific-targeted public to be able to read it and grasp continuity of the data. The authors would also prefer to keep the colour ramps consistent between the Northern Hemisphere and this paper.

**Comment:** P24, L10. Is part of the apparent lapse rate behavior that you have observed a function of changing substrate? In low elevation areas, tills and other soils may dominate, while at high elevations, ice- free areas are largely bedrock. How might this affect the apparent MAGT lapse rate?

Response: The observed lapse behaviour is not a function of changing substrates since this information was unfortunately not included in the model. Due to the low TDDs the substrate properties taken into account by differences in the rk-factors would not have much influence on the modelled results, especially in the coldest parts of the Antarctic.

**Comment:** P24, L14. The increase in MAGT with elevation at some sites is extremely curious. What do you attribute it to? Is this explained in P30, L25? It might be worth highlighting this curious result sooner.

Response: It is difficult to attribute this lapse rate behaviour to any specific phenomenon. The ice-free areas of similar elevation are scattered around different areas inside regions, which have differing continentality and latitude characteristics. The differences are, in our opinion, more likely to cause the MAGT increase with elevation than the wide-spread temperature inversions. The effect of the temperature inversions, however, cannot be excluded.

Changes: The following sentence was added: "The MAGT increase with elevation could be explained by presence of rock outcrops with similar elevation in different latitudes or in different settings regarding continentality." **Comment:** P27, L4. Given the importance of orientation and insolation in controlling ground surface temperature in Antarctic ice free areas, would it be worth adding an orientation factor to the model? Something like fraction of north-facing surfaces in the grid cell? With datasets like REMA now available, it might significantly reduce the offsets in these numerous "microclimate" sites?

Response: The authors already considered including microclimate differences due to different terrain exposition in the model for the Northern Hemisphere. Although simple, the TTOP model is physically based and integration of MAGST relationships with terrain and slope aspect would require establishment of physical relationships. These physical relationships are in respect to the TTOP model not known thus we decided not to include this relationship. An additional challenge are the input MODIS data, which are acquired vertically, so the temperature signal from vertical slopes is not captured.

**Comment:** P31, L5. An RMSE of 1.9°C is really pretty good, but it could be a substantial uncertainty for warmer permafrost sites. How does uncertainty scale with absolute or mean average temperature? Are there particularly warm sites that have high uncertainty? How might the distribution of uncertainty affect our understanding of which permafrost is at risk of thaw?

Response: The TTOP model is an equilibrium model and does not take in to account latent heat released upon thawing of ice-rich permafrost or delayed freezing under cooling conditions. MAGTs at the warm permafrost sites on the Antarctic Peninsula have mostly been underestimated, but in most cases the difference is less than 2 °C. The greatest uncertainties occur on nunataks where the snow cover has been considerably overestimated.

According to the validation data the uncertainties are not the greatest at warm permafrost sites, although higher uncertainties can be expected on the ice-rich sites that are subject to rapid permafrost warming. A transient permafrost model would have to be used to predict which permafrost is at risk of thaw.

LST measurements in the gap-filled temperature dataset can be considered as a measure for the input satellite data.

Changes: The percentage of MODIS LST measurements in the combined MODIS/ERA surface temperature product are now included in Figure 1.

**Comment:** 2. Figures can be improved by proper color choice following other referee's suggestions. Also an overall figure of whole Antarctica with the modelled MAGT should be added at the end to give a general idea of MAGT in the whole continent to guide future studies. Forcing data uncertainty can be added to that figure.

Response: The authors had to make a compromise between how intuitive the colour ramps are and how much information can be put on the maps. Using intuitive colour ramps with one or two colours would make impossible to distinguish the number of the temperature classes that are presented on the maps. To be consistent between the studies, we prefer to keep the colour ramps the same as those that were used for the Northern Hemisphere by Obu et al. (2019). We also believe that the colour scheme is a good compromise between putting as much information as possible on the map and the ability for a scientific-targeted public to be able to read it and grasp continuity of the data.

The scale of the rock outcrops is so small in respect to the ice areas, that it is cartographically impossible to show a raster dataset at an Antarctic-wide scale. Thus, several maps had to be created at a regional scale.

**Comment:** 3. The model limitations should also include possible sources of uncertainty for nf,nt, and rk values such as different thermal properties, soil hydrological conditions etc. Also it would be helpful to mention Cryogrid's difference to process-based models that are mainly used for northern hemisphere permafrost studies.

Response: The model limitations according to the nf-factors were described in detail in section 4.3. We omitted nt factors in the TTOP model since we use land surface temperature instead of air temperature. Discussion about ground properties and rk-factors is extended.

Changes: A whole paragraph was added (fourth paragraph in the new version) in section 4.3 regarding the different soil thermal properties and rk-factors. A transient permafrost model was mentioned at the end of this paragraph.

**Minor comments**

**Comment:** - fig2: correct the mismatch MAE written on figure (-0.17) and in text (0.06)

Changes: Changed accordingly to -0.17 in the text

Comment: - stay consistent with abbreviations, MAGTs vs MAGTS (p8116) -

Changes: Changed accordingly to MAGTs, which is now plural of MAGT.

**Comment:** p2417 correct "The lapse rates are is indicating: : :"

Changes: Changed accordingly to "The lapse rates indicate..."

Comment: - for section 4, please add corresponding figure numbers when discussing specific sites

Changes: Changed accordingly.

**Comment:** - conclusion at p31lines1-3, snow cover and redistribution effects should be added as an important factor as seen from the validations of this paper

Response: We agree that snow cover is an important factor governing local scale permafrost temperature distribution.

Changes: Following sentence was added: "Snow cover, and snow redistribution have strong influence local permafrost temperature variations in the Antarctic"

[revised manuscript text omitted]